# IBMEA: Exploring Variational Information Bottleneck for Multi-modal Entity Alignment

### Taoyu Su
Institute of Information Engineering,
Chinese Academy of Sciences
School of Cyber Security, University
of Chinese Academy of Sciences
Beijing, China
sutaoyu@iie.ac.cn

### Jiawei Sheng*
Institute of Information Engineering,
Chinese Academy of Sciences
Beijing, China
shengjiawei@iie.ac.cn

### Shicheng Wang
Institute of Information Engineering,
Chinese Academy of Sciences
School of Cyber Security, University
of Chinese Academy of Sciences
Beijing, China
wangshicheng@iie.ac.cn

### Xinghua Zhang
Institute of Information Engineering,
Chinese Academy of Sciences
School of Cyber Security, University
of Chinese Academy of Sciences
Beijing, China
zhangxinghua@iie.ac.cn

### Hongbo Xu
Institute of Information Engineering,
Chinese Academy of Sciences
Beijing, China
hbxu@iie.ac.cn

### Tingwen Liu
Institute of Information Engineering,
Chinese Academy of Sciences
School of Cyber Security, University
of Chinese Academy of Sciences
Beijing, China
liutingwen@iie.ac.cn

## Abstract

Multi-modal entity alignment (MMEA) aims to identify equivalent entities between multi-modal knowledge graphs (MMKGs), where entities can be associated with related images. Most existing studies rely heavily on the automatically learned multi-modal fusion modules, which may allow redundant information such as misleading clues in the generated entity representations, impeding the feature consistency of equivalent entities. To this end, we propose a variational framework for MMEA via information bottleneck, termed as IBMEA, by emphasizing alignment-relevant information while suppressing alignment-irrelevant information in entity representations. Specifically, we first develop multi-modal variational encoders that represent modal-specific features as probability distributions. Then, we propose four modal-specific information bottleneck regularizers to limit the misleading clues in the modal-specific entity representations. Finally, we propose a modal-hybrid information contrastive regularizer to integrate modal-specific representations and ensure the similarity of equivalent entities between MMKGs to achieve MMEA. We conduct extensive experiments on 2 cross-KG and 3 bilingual MMEA datasets. Experimental results demonstrate that our model consistently outperforms previous state-of-the-art methods, and also shows promising and robust performance especially in the low-resource and high-noise data scenarios.

## CCS Concepts

• **Information systems → Data mining**; **Multimedia information systems**; **Information integration**;

---

*Corresponding author.

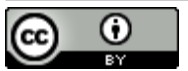

*MM '24, October 28-November 1, 2024, Melbourne, VIC, Australia*
© 2024 Copyright held by the owner/author(s).
ACM ISBN 979-8-4007-0686-8/24/10
https://doi.org/10.1145/3664647.3680954

## Keywords

Multi-modal entity alignment, Multi-modal knowledge graph, Information bottleneck

### ACM Reference Format:

Taoyu Su, Jiawei Sheng, Shicheng Wang, Xinghua Zhang, Hongbo Xu, and Tingwen Liu. 2024. IBMEA: Exploring Variational Information Bottleneck for Multi-modal Entity Alignment. In *Proceedings of the 32nd ACM International Conference on Multimedia (MM '24), October 28-November 1, 2024, Melbourne, VIC, Australia.* ACM, New York, NY, USA, 10 pages. https://doi.org/10.1145/3664647.3680954

## 1 Introduction

Recent years have witnessed the booming of *multi-modal knowledge graphs (MMKGs)*, which extend traditional *knowledge graphs (KGs)* by introducing multi-modal data, such as the relevant images of entities, to provide physical world meanings to the symbols of KGs. Along this line, various downstream applications can be supported, such as visual question answering [52], recommendation systems [9, 48] and other applications [20, 50, 53]. However, MMKGs are usually constructed from separate multi-modal resources for different purposes, often suffering from the issue of incompleteness and limited coverage [24]. Therefore, the task of *multi-modal entity alignment (MMEA)* has been proposed, which aims to identify equivalent entities between two MMKGs, by accessing the multi-modal information of entities, such as their structures, relations, attributes, and images. In this way, one MMKG can retrieve and acquire useful knowledge from other MMKGs.

To achieve MMEA, a prominent challenge is how to exploit the feature consistency of equivalent entities between MMKGs from their diverse and abundant multi-modal information. Pioneer methods such as PoE [24] concatenate all modality features to generate holistic multi-modal entity representations, but neglect the diverse importance of different modality features. Subsequently, effective methods [3, 13, 22, 23] attempt sophisticated multi-modal fusion

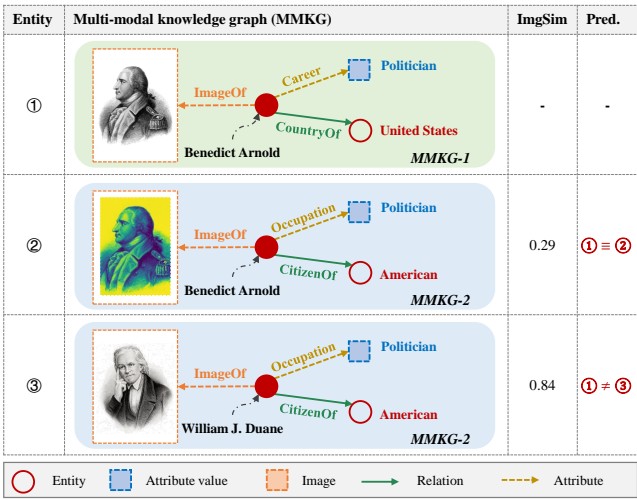

| Entity | Multi-modal knowledge graph (MMKG) | ImgSim | Pred. |
|---|---|---|---|
| ① | Benedict Arnold — ImageOf, Career → Politician, CountryOf → United States, *MMKG-1* | - | - |
| ② | Benedict Arnold — ImageOf, Occupation → Politician, CitizenOf → American, *MMKG-2* | 0.29 | ① ≡ ② |
| ③ | William J. Duane — ImageOf, Occupation → Politician, CitizenOf → American, *MMKG-2* | 0.84 | ① ≠ ③ |

○ Entity  ▢ Attribute value  ▢ Image  → Relation  ----→ Attribute

**Figure 1: An example of the MMEA task between MMKGs, where ImgSim denotes the similarity of the images. Given `Entity_1` in *MMKG-1*, the model aims to predict `Entity_2` from candidate entities in *MMKG-2* as the true entity.**

modules to focus more on the important modality features, benefiting to emphasize consistent features for equivalent entities. For example, EVA [23] learns global attention weights for different modalities, and MEAformer [3] further develops dynamic modality weights specifically for different entities. However, these methods rely heavily on the automatically learned fusion module, which can be hampered by the alignment-irrelevant information such as *misleading clues*[1] in the entities' multi-modal information, especially when there are only low-resource or high-noise training data.

In contrast, we argue that it is crucial to consider the misleading clues in modalities for MMEA. As shown in Figure 1, given `Entity_1` in *MMKG-1*, we are going to search the candidate entities (e.g., `Entity_2`, `Entity_3`, and so on) from *MMKG-2*, and identify `Entity_2` as the true equivalent entity. However, we accidentally find that the image of `Entity_1` has higher similarity[2] with `Entity_3` due to the similar background color. Even though, we can still identify `Entity_1` and `Entity_2` as the equivalent entities by recognizing the details of human faces. Actually, in this case, the background color reflects the misleading clue that is alignment-irrelevant for MMEA. Existing methods usually struggle[3] with this case since the multi-modal fusion modules may hardly discern the alignment-relevant information from the entire image information. This inspires us to emphasize the alignment-relevant information explicitly, and simultaneously surpass the alignment-irrelevant information in modalities, to relieve the impact of misleading clues.

For this purpose, we explore *information bottleneck (IB)* [35, 36] as a potential solution. In general, the IB principle aims to learn an ideal representation that makes a trade-off between the fully data descriptive information and the task predictive information [36].

---

[1]It is also usually called shortcuts in researches [14].
[2]Note that we select examples from real datasets, and obtain image features from ResNet-152 to calculate cosine similarity. For details, please see Sec. 4.4.2.
[3]For details, we show empirical results in Figure. 5, 6, and 7.

For the MMEA task, we expect the multi-modal encoders of different modalities to extract alignment-relevant features as the task predictive information, while appropriately ignoring certain alignment-irrelevant information in the data descriptive information. In this way, the model is required to focus more on the necessary information for MMEA from the contexts of entity pairs between MMKGs, thus limiting the redundant information in entity representations and improving the robustness to misleading clues.

Following the above idea, we propose a novel variational framework for **M**ulti-modal **E**ntity **A**lignment via **I**nformation **B**ottleneck, termed as **IBMEA**. Specifically, we first devise multi-modal variational encoders to capture the multi-modal features of each entity, including its graph structures, images, relations and attributes, and then represent them with probability distributions. Afterward, we propose two kinds of multi-modal information regularizers: 1) the modal-specific information bottleneck regularizer, which takes each modal-specific representation as input, surpassing alignment-irrelevant information and emphasizing alignment-relevant information, 2) the modal-hybrid information contrastive regularizer, which adaptively fuses all modal-specific representations to generate modal-hybrid representations, and further enhances entity similarity between entity pairs to achieve MMEA. Our major contributions can be summarized as follows:

- We introduce a novel perspective for MMEA with IB, and propose a variational framework as IBMEA. To our knowledge, we are the first to explore IB to alleviate misleading clues in MMEA.
- We propose two kinds of regularizers to refine the modal-specific and modal-hybrid features, which surpass alignment-irrelevant information and emphasize alignment-relevant information, improving the robustness with misleading clues.
- Experiments indicate that our model outperforms the comparison state-of-the-art methods on 5 benchmarks, and obtains promising and robust results in low-resource and high-noise scenarios.

## 2 Preliminaries

### 2.1 Task Formulation

Formally, the ***multi-modal knowledge graph (MMKG)*** can be defined as $\mathcal{G} = (\mathcal{E}, \mathcal{R}, \mathcal{A}, \mathcal{V})$, where $\mathcal{E}, \mathcal{R}, \mathcal{A}, \mathcal{V}$ are the sets of entities, relations, attributes and visual images, respectively. Therefore, the triples are defined as $\mathcal{T} \subseteq \mathcal{E} \times \mathcal{R} \times \mathcal{E}$, where each entity $e \in \mathcal{E}$ can be linked to attributes and images. Building on previous research [3, 22, 23], we mainly focus four modalities $m \in \{g, v, r, a\}$ of entities, including graph structures $g$, visual images $v$, neighboring relations $r$ and attributes $a$.

Based upon, ***multi-modal entity alignment (MMEA)*** aims to identify equivalent entities from two different MMKGs. Formally, given two MMKGs $\mathcal{G}^{(1)}$ and $\mathcal{G}^{(2)}$ with their relational triples and multi-modal attributes, the goal of MMEA task is to identify equivalent entity pairs $\{\langle e^{(1)}, e^{(2)} \rangle | e^{(1)} \in \mathcal{G}^{(1)}, e^{(2)} \in \mathcal{G}^{(2)}, e^{(1)} \equiv e^{(2)}\}$. During training, the model utilizes a set of pre-aligned entity pairs $\mathcal{S}$ as alignment seeds, while testing focuses on predicting equivalent entity pairs across MMKGs. In this paper, we aim to achieve MMEA by learning entity representations from both $\mathcal{G}^{(1)}$ and $\mathcal{G}^{(2)}$, emphasizing alignment-relevant information while surpassing alignment-irrelevant information to bridge between MMKGs.

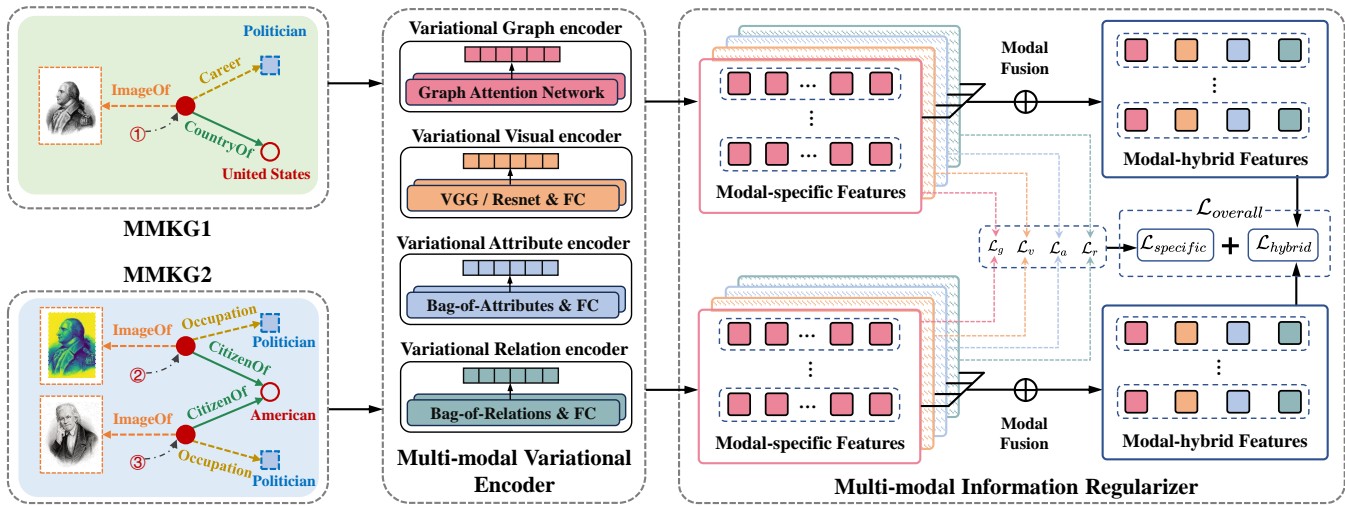

**Figure 2: The framework of the proposed IBMEA for the multi-modal entity alignment task.**

## 2.2 Information Bottleneck

We adopt the information bottleneck (IB) principle [35, 36] to learn general entity representations across MMKGs. The IB principle aims to find a maximally compressed representation $Z$ of the input $X$ while preserving information about the target $Y$, which achieved by minimizing the objective function:

$$\mathcal{L}_{IB} = \beta I(Z; X) - I(Z; Y), \qquad (1)$$

where $I(\cdot; \cdot)$ denotes mutual information, and $\beta > 0$ is a Lagrangian multiplier controlling the trade-off between compression and information preservation. Minimizing the first term penalizes the information between $Z$ and $X$, encouraging the latent variable $Z$ to forget irrelevant information. Maximizing the second term encourages $Z$ to be predictive of $Y$. Consequently, the IB principle enables $Z$ to capture relevant factors for prediction while compressing irrelevant parts [6]. This aligns with the notion of $Z$ acting as a minimal sufficient statistic of $X$ for predicting $Y$ [5].

Although the IB principle is appealing, computing mutual information poses computational challenges. To overcome this, the VIB (Variational Information Bottleneck) [5] method proposes a variational approximation approach for the IB objective. It minimizes the following formulation:

$$\mathcal{L}_{\text{VIB}} = \beta \, \mathbb{E}_{\boldsymbol{x}} \left[ \text{KL} \left[ p_\theta(\boldsymbol{z}|\boldsymbol{x}) || r(\boldsymbol{z}) \right] \right] - \mathbb{E}_{\boldsymbol{z} \sim p_\theta(\boldsymbol{z}|\boldsymbol{x})} \left[ \log q_\phi(\boldsymbol{y}|\boldsymbol{z}) \right], \quad (2)$$

where $p_\theta(\boldsymbol{z}|\boldsymbol{x})$ is an estimate of posterior probability to generate $\boldsymbol{z}$ from data $\boldsymbol{x}$, $r(\boldsymbol{z})$ is an estimate of the prior probability $p(\boldsymbol{z})$ to describe $\boldsymbol{z}$, and $q_\phi(\boldsymbol{y}|\boldsymbol{z})$ is a parametric approximation of $p(\boldsymbol{y}|\boldsymbol{z})$ to predict target $\boldsymbol{y}$. Intuitively, the encoder generates $p_\theta(\boldsymbol{z}|\boldsymbol{x})$ with parameter $\theta$, and the decoder achieves $q_\phi(\boldsymbol{y}|\boldsymbol{z})$ with parameter $\phi$. Actually, Eq. (2) resembles a likelihood-prior trade-off, since the first term applies KL-divergence with uninformative prior, penalizing the too complex encoded representations, and the second term approximates the log-likelihood expectation, requiring the representations to achieve task prediction.

## 3 Methodology

The overall framework of IBMEA is illustrated in Figure 2. Our model mainly consists of two components: 1) *Multi-modal Variational Encoder*, which represents multi-modal information of entities from both MMKGs into random variables, and 2) *Multi-modal Information Regularizer*, which imposes IB regularization on each modal-specific representations, and facilitates MMEA with contrastive regularization upon modal-hybrid representations.

## 3.1 Multi-modal Variational Encoder

To capture entity features from all modalities, including the graph structure $g$, visual image $v$, relation $r$ and attribute $a$. We devise the modal-specific variational encoder and generate the modal-specific feature variables as $\boldsymbol{z}_m$ with $m \in \{g, v, r, a\}$, respectively.

*3.1.1 Variational Graph Encoder.* To fully leverage the graph structures of MMKGs, we devise a variational graph encoder. Considering the neighbors of an entity can have different importance for entity alignment, we adopt (two attention heads and two layers) graph attention network (GAT) [40] to capture graph structures. To better depict the information of the graph structural feature, we represent it with a variable $\boldsymbol{z}_g$ of the probability distribution. Practically, we assume the variable as Gaussian distributions as widely used in variational studies [5, 16, 44], and represent $\boldsymbol{z}_g$ with mean vector $\boldsymbol{\mu}_g$ and variance vector $\boldsymbol{\sigma}_g$, derived by separate GATs with different learnable parameters [16]:

$$\begin{aligned} \boldsymbol{\mu}_g &= \text{GAT}(\boldsymbol{A}, \boldsymbol{x}_g; \theta_{g,\mu}), \\ \boldsymbol{\sigma}_g &= \text{GAT}(\boldsymbol{A}, \boldsymbol{x}_g; \theta_{g,\sigma}), \\ \boldsymbol{z}_g &\sim \mathcal{N}(\boldsymbol{\mu}_g, \text{diag}(\boldsymbol{\sigma}_g^2)), \end{aligned} \qquad (3)$$

where $\boldsymbol{x}_g \in \mathbb{R}^{d_g}$ is the randomly initialized node features, $\boldsymbol{A}$ denotes the adjacent matrix of the given MMKG. To alleviate the negative impacts of structural heterogeneity, we encode both MMKGs with shared GATs to project entities into the same embedding space [18]. The learnable parameters can be summarized as $\theta_g \triangleq \{\theta_{g,\mu}, \theta_{g,\sigma}\}$.

### 3.1.2 Variational Visual, Attribute and Relation Encoder.
To highlight the information of relations, attributes, and images, we devise separate fully connected layers as multi-modal encoders to learn interim representations for each modality $m' \in \{v, a, r\}$ as follows:

$$\widehat{h}_{m'} = \delta(\text{FC}_{m'}(W_{m'}, x_{m'})), \tag{4}$$

where $W_{m'} \in \mathbb{R}^{d_{m'} \times d}$ and $\delta$ is the ReLU activation. Here, $x_{m'} \in \mathbb{R}^{d_{m'}}$ denotes the initialized feature for $m'$-modality. For images, we utilize pre-trained visual encoders to obtain effective image features. For entities without images, we average the images of their neighbors as the initial feature [19]. For attributes and relations, the bag-of-attributes and bag-of-relations features [49] are employed.

Based on these interim representations, we employ multi-layer perceptrons (MLPs) to derive modal-specific feature variables. Assuming these variables follow a Gaussian distribution and employ separate MLPs to learn mean vector $\mu_{m'}$ and variance vector $\sigma_{m'}$:

$$\begin{aligned} \mu_{m'} &= \text{MLP}(\widehat{h}_{m'}; \theta_{m',\mu}), \\ \sigma_{m'} &= \text{MLP}(\widehat{h}_{m'}; \theta_{m',\sigma}), \\ z_{m'} &\sim \mathcal{N}(\mu_{m'}, \text{diag}(\sigma_{m'}^2)), \end{aligned} \tag{5}$$

where the learnable parameters for $m'$-modality can be summarized as $\theta_{m'} \triangleq \{\theta_{m',\mu}, \theta_{m',\sigma}, W_{m'}\}$ with $m' \in \{v, a, r\}$.

### 3.1.3 Multi-modal Representation Implementation.
Since the modal-specific variables are intractable for neural networks, we leverage the reparameterization trick [16] to sample deterministic representations from probability distributions. For each modality $m \in \{g, v, a, r\}$, the sampled representation $z_m$ is obtained as follows:

$$z_m = \mu_m + \sigma_m \odot \epsilon, \epsilon \sim \mathcal{N}(0, \text{diag}(I)), \tag{6}$$

where $\mu_m$ and $\sigma_m$ are the corresponding mean and standard deviation of $m$-modality, respectively. $\epsilon$ is a standard Gaussian noise, and $\odot$ denotes the element-wise production. In this way, we obtain the deterministic modal-specific representations for entities, which can be further integrated to acquire entire (modal-hybrid) entity representations to make predictions for MMEA.

## 3.2 Multi-modal Information Regularizer

In this section, we introduce two kinds of regularizers from mutual information perspective, which refine the modal-specific representations with limited alignment-irrelevant information, and derive better modal-hybrid representations for MMEA.

### 3.2.1 Modal-specific Information Bottleneck Regularizer.
As claimed in Sec. 1, there may exist alignment-irrelevant misleading clues in distinct modalities. For each modality $m \in \{g, v, a, r\}$, we expect the learned modal-specific feature variables $z_m$ to contain less alignment-irrelevant information while retaining more alignment-relevant information between MMKGs. Following this idea, we propose the information bottleneck regularizer $\mathcal{L}_m$ for $m$-modality feature variables of the paired entities from the two MMKGs:

$$\begin{aligned} \mathcal{L}_m &= \sum_{i \in \{1,2\}} \left[ \beta_m I(Z_m^{(i)}; X_m^{(i)}) - I(Z_m^{(i)}; Y) \right], \\ &= \underbrace{\beta_m \left[ I(Z_m^{(1)}; X_m^{(1)}) + I(Z_m^{(2)}; X_m^{(2)}) \right]}_{\text{Minimality}} - \underbrace{I(Z_m^{(1)}, Z_m^{(2)}; Y)}_{\text{Alignment}}, \end{aligned} \tag{7}$$

where $X_m^{(1)}$ and $X_m^{(2)}$ denotes the original $m$-modality features from $\mathcal{G}^{(1)}$ and $\mathcal{G}^{(2)}$, respectively. $Z_m^{(1)}$ and $Z_m^{(2)}$ are obtained by Eq. (6) over the two MMKGs. $Y$ denotes the supervised information, indicating whether the paired entities are equivalent. Note that *the minimality term penalizes the information from its own MMKG, while the alignment term encourages the representations to be capable of entity alignment.* In this way, the redundant information is surpassed by minimality term, alleviating alignment-irrelevant misleading clues in the modal-specific representations. On the other hand, the alignment-relevant information is simultaneously emphasized, since we require the modal to make prediction by the alignment term with tailored information. For tractable objective function, please refer to Sec. 3.3.1.

To refine modal-specific representations of all modalities, the overall IB regularizer $\mathcal{L}_{specific}$ is defined as follows:

$$\mathcal{L}_{specific} = \sum_{m \in \{g,v,a,r\}} \mathcal{L}_m. \tag{8}$$

In this way, we encourage each modality of information to remain consistency between MMKGs, benefiting to entire representations.

### 3.2.2 Modal-hybrid Information Contrastive Regularizer.
To fully leverage information of all modalities for MMEA, we propose a modal-hybrid information contrastive regularizer. We first generate modal-hybrid feature variables by integrating all the modal-specific feature variables. Specifically, for each entity, we measure the distinct importance of its modality information with attention mechanism, and employ the attention weights to integrate modal-specific feature variables (sampled from Eq. (6)) as follows:

$$\begin{aligned} s_m &= q_o^T \tanh(W_m z_m + b_m), \\ \alpha_m &= \frac{\exp(s_m)}{\sum_{k \in \{g,v,a,r\}} \exp(s_k)}, \\ z_o &= \sum_{m \in \{g,v,a,r\}} \alpha_m z_m, \end{aligned} \tag{9}$$

where $\alpha_m$ is the attention weight for modality $m$, taking the different nature of entities into consideration. In addition, $q_o \in \mathbb{R}^d$, $W_m \in \mathbb{R}^{d \times d_m}$ and $b_m \in \mathbb{R}^d$ are learnable parameters. In this way, we obtain modal-hybrid feature variables considering the distinct modality importance of the entity and leverage the IB-refined modal-specific feature variables.

Based on the derived modal-hybrid feature variable, we aim to measure the mutual information between paired entities from the two MMKGs. Our main intuition is that, *given an entity in the one MMKG, the equivalent entity in the other MMKG would have higher mutual information than other candidate entities.* Following this intuition, we propose the modal-hybrid information contrastive regularizer $\mathcal{L}_{hybrid}$ as follows:

$$\mathcal{L}_{hybrid} = -\underbrace{I(Z_o^{(1)}; Z_o^{(2)})}_{\text{Contrastive term}}. \tag{10}$$

In this way, we measure the consistency between paired entities from the two MMKGs, which fully leverages all the IB-refined modal-specific features to constitute the entire entity representations. For tractable objective function, please refer to Sec. 3.3.2.

## 3.3 Tractable Optimization Objective

Theoretically, direct optimization of mutual information can be intractable considering the intractable integrals [6, 29]. Therefore, we derive tractable objectives with variational approximation [38].

### 3.3.1 Tractable Information Bottleneck Objective.
To optimize the information bottleneck regularizer in Eq. (7), we derive the variational lower bound for the minimality and alignment term.

For the **minimality term**, we measure it by Kullback-Leibler (KL) divergence [4] with variational approximation posterior distributions. Specifically, recall that we are given the original feature $x_m$ of $m$-modality from $\mathcal{G}$ ($m \in \{g, v, r, a\}$, and we omit the subscript of different MMKGs), and the feature variable $z_m$ can be approximated by the $m$-th modality variational encoder. We can derive the approximation upper bound [29] of the minimality term, and minimize it as the tractable objective function:

$$
\begin{aligned}
I(Z_m; X_m) &\leq \mathbb{D}_{KL}(p_\theta(z_m|x_m)||r(z_m)) \\
&= \mathbb{D}_{KL}(\mathcal{N}(\boldsymbol{\mu}_m, \text{diag}(\boldsymbol{\sigma}_m^2))||\mathcal{N}(0, \text{diag}(\boldsymbol{I}))).
\end{aligned}
\tag{11}
$$

Here, to approximate true posterior distributions $p(z_m|x_m)$, we adopt the $m$-modality variational encoder parameterized by $\theta$ (omit subscript $m$ of $\theta_m$) to generate approximated posterior distribution as $p_\theta(z_m|x_m)$. Besides, following existing variational methods [5, 16, 44], we also assume the prior distribution $r(z_m)$ as a standard Gaussian distribution $\mathcal{N}(0, \text{diag}(\boldsymbol{I}))$ for convenience. Such a procedure holds for both MMKG $\mathcal{G}^{(1)}$ and $\mathcal{G}^{(2)}$. In this way, we compress redundant information in the $m$-modality representation.

For the **alignment term**, we also adopt variational approximation posterior distribution to render. The goal of the alignment term is to make prediction with the learned feature variables. Therefore, we rewrite the alignment term with likelihood of the entity alignment task, which is the variational lower bound for maximization as the tractable objective function:

$$
\begin{aligned}
&I(Z_m^{(1)}, Z_m^{(2)}; Y) \\
&\geq \mathbb{E}_{z_m^{(1)}, z_m^{(2)} \sim p_\theta(z_m^{(1)}|x_m^{(1)}), p_\theta(z_m^{(2)}|x_m^{(2)})}[\log q_\phi(y|z_m^{(1)}, z_m^{(2)})],
\end{aligned}
\tag{12}
$$

where the variable $y$ denotes whether the entity pair is equivalent, and $q_\phi(y|z_m^{(1)}, z_m^{(2)})$ is the entity alignment decoder to make prediction. Generally, the likelihood can be achieved by binary cross-entropy, and here we use InfoNCE [39] to better consider all the candidate entities in ranking, which can be formed as:

$$
\begin{aligned}
I(Z_m^{(1)}, Z_m^{(2)}; Y) \geq &\sum_{(z_m^{(1)}, z_m^{(2)}) \in \mathcal{S}} \big[ \log \exp(\cos(z_m^{(1)}, z_m^{(2)})/\tau) \\
&- \sum_{(z_m^{(1)}, \hat{z}_m^{(2)}) \notin \mathcal{S}} \log \exp(\cos(z_m^{(1)}, \hat{z}_m^{(2)})/\tau) \big],
\end{aligned}
\tag{13}
$$

where $\mathcal{S}$ is the seed alignments (i.e., pre-aligned entity pairs), $\tau$ is the temperature factor. $\hat{z}_m^{(2)}$ is the negative entity obtained by randomly replacing the true entity of seed alignments.

### 3.3.2 Tractable Information Contrastive Objective.
To optimize the contrastive term, we follow the idea of InfoMax [41] by measuring the mutual information with neural networks. Specifically, we use another discriminator D to measure the consistency between entities, and the lower bound of the contrastive term can be derived as:

$$
\begin{aligned}
I(Z_o^{(1)}; Z_o^{(2)}) &= \mathbb{E}_{p(z_o^{(1)}|x^{(1)})p(z_o^{(2)}|x^{(2)})}[\log D(z_o^{(1)}, z_o^{(2)})] \\
&\geq \sum_{(z_o^{(1)}, z_o^{(2)}) \in \mathcal{S}} \log(D(z_o^{(1)}, z_o^{(2)})) + \sum_{(z_o^{(1)}, z_o^{(2)}) \notin \mathcal{S}} \log(1 - D(z_o^{(1)}, z_o^{(2)})),
\end{aligned}
\tag{14}
$$

where $z_o^{(1)}$ and $z_o^{(2)}$ are modal-hybrid variables for $\mathcal{G}^{(1)}$ and $\mathcal{G}^{(2)}$ from Eq. (9) respectively, which encode all the original modality features ($x \triangleq \{x_g, x_v, x_a, x_r\}$) in the corresponding MMKG. Here we can achieve the discriminator D with inner production followed by sigmoid. Inspired by Boudiaf et al. [8], we adopt multi-similarity loss [43] to further consider the impact of different entities in contrastive learning. In this way, we encourage the modal-hybrid representations of the equivalent entities to be relevant and thereupon achieve the MMEA task with all modality information.

### 3.3.3 Overall Objective Function.
Since we have imposed constraints on both the modal-specific and modal-hybrid representations, the overall objective function is defined in a joint paradigm as follows:

$$
\mathcal{L}_{overall} = \mathcal{L}_{specific} + \mathcal{L}_{hybrid}.
\tag{15}
$$

In summary, the proposed IB regularizers enable us to suppress alignment-irrelevant information and emphasize the alignment-relevant information to fulfill the MMEA task.

## 4 Experiments

In this section, we conduct extensive experiments to evaluate the effectiveness. More analyses are detailed in **Appendix**[4].

### 4.1 Experimental Settings

#### 4.1.1 Datasets and Evaluation Metrics.
In this study, we evaluate our proposed IBMEA on five multi-modal EA datasets, categorized into two types. (1) Cross-KG [24] datasets, comprising **FB15K-DB15K** and **FB15K-YAGO15K**, with 128,486 and 11,199 labeled pre-aligned entity pairs, respectively. (2) Bilingual datasets also named **DBP15K** [23, 32], which are including **DBP15K_ZH-EN**, **DBP15K_JA-EN** and **DBP15K_FR-EN** from the multilingual versions of DBpedia, each with approximately 400K triples and 15K pre-aligned entity pairs. EVA [23] provided images of entities for the DBP15K dataset. Following previous works [3, 22, 23], we utilize 20%, 50%, 80% of true entity pairs as alignment seeds for training on cross-KG datasets, and 30% for bilingual datasets. We use Hits@1 (H@1), Hits@10 (H@10), and Mean Reciprocal Rank (MRR) as evaluation metrics. Hits@N denotes the proportion of correct entities in the top N ranks, while MRR is the average reciprocal rank. For both metrics, higher values indicate better alignment results.

#### 4.1.2 Baselines.
To verify the effectiveness of IBMEA, we select several typical and competitive methods as baselines. We manifest these methods into two groups: **Traditional Entity Alignment methods.** We choose 6 prominent EA methods proposed in recent years, which achieve entity alignment based on the graph structures and do not introduce multi-modal information, including TransE [7], IPTransE [54], GCN-align [45], KECG [18], BootEA [33], and NAEA [55]. **MMEA methods.** We further collect 10 state-of-the-art MMEA methods, which incorporate entity images as input features to enrich entity representations, including POE [24], Chen

---

[4]Our **Appendix** and **code** are available at https://github.com/sutaoyu/IBMEA.

**Table 1: Experimental results on 2 cross-KG datasets where X% represents the percentage of seed alignments used for training. The best result is bold-faced and the runner-up is underlined. ∗ indicates results reproduced using the official source code.**

| Methods | FB15K-DB15K (20%) | | | FB15K-DB15K (50%) | | | FB15K-DB15K (80%) | | | FB15K-YAGO15K (20%) | | | FB15K-YAGO15K (50%) | | | FB15K-YAGO15K (80%) | | |
|---|---|---|---|---|---|---|---|---|---|---|---|---|---|---|---|---|---|---|
| | H@1 | H@10 | MRR | H@1 | H@10 | MRR | H@1 | H@10 | MRR | H@1 | H@10 | MRR | H@1 | H@10 | MRR | H@1 | H@10 | MRR |
| TransE [7] | .078 | .240 | .134 | .230 | .446 | .306 | .426 | .659 | .507 | .064 | .203 | .112 | .197 | .382 | .262 | .392 | .595 | .463 |
| IPTransE [54] | .065 | .215 | .094 | .210 | .421 | .283 | .403 | .627 | .469 | .047 | .169 | .084 | .201 | .369 | .248 | .401 | .602 | .458 |
| GCN-align [45] | .053 | .174 | .087 | .226 | .435 | .293 | .414 | .635 | .472 | .081 | .235 | .153 | .235 | .424 | .294 | .406 | .643 | .477 |
| KECG* [18] | .128 | .340 | .200 | .167 | .416 | .251 | .235 | .532 | .336 | .094 | .274 | .154 | .167 | .381 | .241 | .241 | .501 | .329 |
| POE [24] | .126 | .151 | .170 | .464 | .658 | .533 | .666 | .820 | .721 | .113 | .229 | .154 | .347 | .536 | .414 | .573 | .746 | .635 |
| Chen et al. [10] | .265 | .541 | .357 | .417 | .703 | .512 | .590 | .869 | .685 | .234 | .480 | .317 | .403 | .645 | .486 | .598 | .839 | .682 |
| HMEA [15] | .127 | .369 | - | .262 | .581 | - | .417 | .786 | - | .105 | .313 | - | .265 | .581 | - | .433 | .801 | - |
| EVA [23] | .134 | .338 | .201 | .223 | .471 | .307 | .370 | .585 | .444 | .098 | .276 | .158 | .240 | .477 | .321 | .394 | .613 | .471 |
| MSNEA [11] | .114 | .296 | .175 | .288 | .590 | .388 | .518 | .779 | .613 | .103 | .249 | .153 | .320 | .589 | .413 | .531 | .778 | .620 |
| ACK-MMEA [19] | .304 | .549 | .387 | .560 | .736 | .624 | .682 | .874 | .752 | .289 | .496 | .360 | .535 | .699 | .593 | .676 | .864 | .744 |
| UMAEA* [13] | .560 | .719 | .617 | .701 | .801 | .736 | .789 | .866 | .817 | .486 | .642 | .540 | .600 | .726 | .644 | .695 | .798 | .732 |
| MCLEA [22] | .445 | .705 | .534 | .573 | .800 | .652 | .730 | .883 | .784 | .388 | .641 | .474 | .543 | .759 | .616 | .653 | .835 | .715 |
| MEAformer [3] | .578 | .812 | .661 | .690 | .871 | .755 | .784 | .921 | .834 | .444 | .692 | .529 | .612 | .808 | .682 | .724 | .880 | .783 |
| IBMEA (Ours) | **.631** | **.813** | **.697** | **.742** | **.880** | **.793** | **.821** | **.922** | **.859** | **.521** | **.708** | **.584** | **.655** | **.821** | **.714** | **.751** | **.890** | **.800** |

et al. [10], HMEA [15], EVA [23], ACK-MMEA [19], MSNEA [11], PSNEA [28], UMAEA [13], MCLEA [22], and MEAformer [3]. For more details, please refer to Sec. 5. Among the methods, MCLEA and MEAformer are typical competitive methods.

*4.1.3 Implementation Details.* In our experiments, the GAT has two layers with a hidden size of $d_g = 300$. For visual embeddings, following [3, 22, 23], we leverage a pre-trained VGG-16 model [31] for cross-KG datasets with $d_v = 4096$ and ResNet-152 for bilingual datasets with $d_v = 2048$ to obtain initial features from the penultimate layer. Following [3, 22], the Bag-of-Words method is chosen to encode both attributes ($x_a$) and relations ($x_r$) as fixed-length vectors, where $d_a$ and $d_r$ are both 1000. The graph embedding output is 300, and other modality embeddings are 100. Training is conducted over 1000 epochs with a batch size of 7,500, using AdamW optimizer [26] with a learning rate of 6e-3 and a weight decay of 1e-2. Hyper-parameters $\beta_g, \beta_v, \beta_a, \beta_r$ in Eq. (7) are tune in [1e-4, 1e-3, 1e-2, 1e-1], yielding optimal results at 1e-3, 1e-2, 1e-2, 1e-2, respectively. Consistent with prior studies [3, 22, 23], we adopt an iterative training strategy to overcome the lack of training data and exclude entity names for fair comparison. Our best hyper-parameters are tuned by grid search according to the prediction accuracy of MMEA, detailed in **Appendix**.

## 4.2 Overall Results

The overall average results on cross-KG and bilingual datasets are displayed in Table 1 and Table 2, respectively. From the tables, we have several observations: 1) *Our proposed IBMEA model outperforms all baseline models on 5 benchmarks with different data settings*, including two cross-KG datasets and three bilingual datasets, excelling in all key metrics (H@1, H@10, and MRR), which demonstrates the model's generality. Specifically, under 50% and 80% alignment seed settings on 2 cross-KG datasets, our model achieved an average increase of 4.2% and 3.0% in H@1 scores, and 3.5% and 2.1% in MRR scores, respectively. Moreover, our model still exceeds those high-performing baselines and increases the current SOTA Hits@1 scores from .847/.842/.845 to .859/.856/.864 on ZH-EN/JA-EN/FR-EN datasets with the DBP15K, respectively. 2) *Our model achieves better results in relatively low-resource data scenario.* Compared to the

**Table 2: Experimental results on 3 bilingual datasets . The best result is bold-faced and the runner-up is underlined. ∗ indicates results reproduced using the official source code.**

| Methods | DBP15K$_{ZH-EN}$ | | | DBP15K$_{JA-EN}$ | | | DBP15K$_{FR-EN}$ | | |
|---|---|---|---|---|---|---|---|---|---|
| | H@1 | H@10 | MRR | H@1 | H@10 | MRR | H@1 | H@10 | MRR |
| GCN-align [45] | .434 | .762 | .550 | .427 | .762 | .540 | .411 | .772 | .530 |
| KECG [18] | .478 | .835 | .598 | .490 | .844 | .610 | .486 | .851 | .610 |
| BootEA [33] | .629 | .847 | .703 | .622 | .854 | .701 | .653 | .874 | .731 |
| NAEA [55] | .650 | .867 | .720 | .641 | .873 | .718 | .673 | .894 | .752 |
| EVA [23] | .761 | .907 | .814 | .762 | .913 | .817 | .793 | .942 | .847 |
| MSNEA [11] | .643 | .865 | .719 | .572 | .832 | .660 | .584 | .841 | .671 |
| UMAEA* [13] | .807 | .967 | .867 | .806 | .967 | .832 | .820 | .979 | .881 |
| PSNEA [28] | .816 | .957 | .869 | .819 | .963 | .868 | .844 | .982 | .891 |
| MCLEA [22] | .816 | .948 | .865 | .812 | .952 | .865 | .834 | .975 | .885 |
| MEAformer [3] | .847 | .970 | .892 | .842 | .974 | .892 | .845 | .976 | .894 |
| IBMEA (Ours) | **.859** | **.975** | **.903** | **.856** | **.978** | **.902** | **.864** | **.985** | **.911** |

runner-up method results, our model achieves an average increase of 4.4% in H@1 and 4.0% in MRR on cross-KG datasets with a 20% alignment seed setting. It demonstrates that using the IB principle, the model can better grasp alignment-relevant information, alleviating overfitting on pieces of predictive clues with limited seed alignments. 3) *The MMEA baseline model generally outperforms the EA baseline.* Remarkably, our model utilizing 20% of the seeds surpasses the H@1 performance of the EA baseline at 80% in cross-KG datasets. It demonstrates that introducing multi-modal information can enrich entity information, and significantly help entity alignment with limited seed alignments. All the results demonstrate the effectiveness of our proposed IBMEA, and we will further examine our model on hard data scenarios to show the results in Sec. 4.4.

## 4.3 Ablation Study

To evaluate the unity of all regularizers, we conduct ablation study on FB15K-DB15K dataset. From the Table 3, we can observe that: 1) The removal of any modal-specific IB regularizer consistently leads to significant reductions across all metrics, thereby validating its efficacy for modal-specific IB regularizers. 2) As the proportion of alignment seeds increases, the negative impact of removing the

**Table 3: Ablation study on IB regularizers. G-IB, V-IB, A-IB, and R-IB are modal-specific IB regularizers (Graph, Visual, Attribute, and Relation) for short. Hybrid-IB refers to using an IB regularizer on the representations after fusion.**

| Model | FB15K-DB15K (20%) | | | FB15K-DB15K (50%) | | | FB15K-DB15K (80%) | | |
|---|---|---|---|---|---|---|---|---|---|
| | H@1 | H@10 | MRR | H@1 | H@10 | MRR | H@1 | H@10 | MRR |
| IBMEA | **.631** | **.813** | **.697** | **.742** | **.880** | **.793** | **.821** | **.922** | **.859** |
| *w/o* G-IB | .597 | .780 | .660 | .707 | .852 | .759 | .806 | .903 | .843 |
| *w/o* V-IB | .623 | .785 | .680 | .725 | .862 | .774 | .816 | .908 | .850 |
| *w/o* A-IB | .613 | .782 | .672 | .721 | .858 | .771 | .807 | .898 | .840 |
| *w/o* R-IB | .625 | .799 | .686 | .720 | .862 | .772 | .816 | .914 | .851 |
| Hybrid-IB | .468 | .614 | .518 | .643 | .762 | .686 | .748 | .837 | .780 |

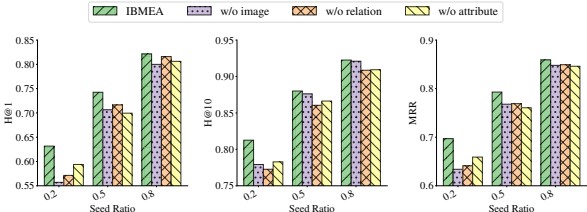

**Figure 3: Results of removing different modalities on FB15K-DB15K dataset. w/o means removing the modality.**

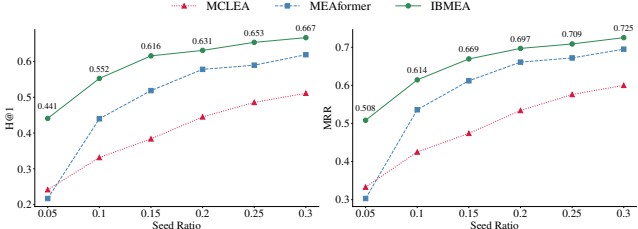

**Figure 4: Results in the low-resource data scenario with proportions of seed alignments on FB15K-DB15K dataset.**

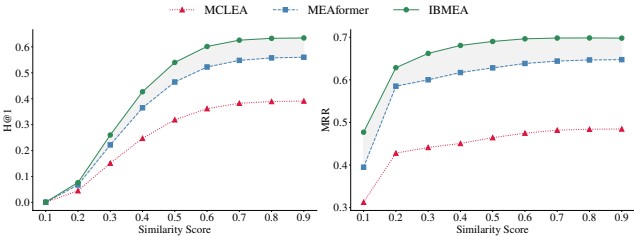

**Figure 5: Results on the samples with low-similarity image in FB15K-DB15K dataset.**

regularizer diminishes. We believe there may exist more misleading information in low-resource settings, thus the modal-specific IB regulariser has a greater effect. 3) Compared with other *w/o* variants, *w/o* G-IB variant exhibits the most dramatic decline in overall results, indicating that the graph structure information plays an important role in MMEA. 4) Comparing hybrid-IB and IBMEA, we see that hybrid-IB results are much lower than IBMEA results, which indicates applying the IB to each modality before fusion is more effective than post-fusion. The IB regularizer used after fusion might indirectly refine essential multi-modal information, thereby hindering the model's capacity to effectively retain alignment-relevant information while suppressing alignment-irrelevant information.

To further evaluate the impact of different modalities, we report the results by deleting different modalities on FB15K-DB15K dataset, shown in Figure 3. We notice that: 1) Removing each modality will reduce the final result, proving each modality's importance. 2) Compared with the high proportion of seeds (50% and 80% ratio), the effect of different modalities information on the overall performance is more obvious in the low proportion of seeds. This indicates that our model can extract crucial information of each modality, thus obtaining better results in the case of the low proportion of seeds.

## 4.4 Further Analysis

*4.4.1 Performance on extreme low-resource training data.* To further explore the performance with few training data, where the alignment seed ratio ranges from 5% to 30%. Specifically, we select MCLEA and MEAformer as baselines, and the results are depicted in Figure 4. We can observe that as the proportion of alignment seeds decreases, the performance of all models would also decrease

in terms of metrics. However, it is obvious that our IBMEA continuously outperforms MCLEA and MEAformer, indicating the effectiveness of our proposed method. Moreover, it is worth noting that the gap between them is much more significant when the seed alignments are extremely few (5%), which confirms the superiority of our proposed method to alleviate the overfitting of shortcut misleading clues in low-resource scenarios.

*4.4.2 Performance on samples with low-similarity image.* To examine the effectiveness of our models on hard samples, we select samples with low-similarity image in FB15K-DB15K dataset (20% seed alignments) and examine the performance of the three MMEA models on these samples. The similarity between the images is calculated by the cosine similarity of the features extracted from the pre-trained visual encoder. Considering alignment entity pairs with low-similarity image, they tend to contain more task-irrelevant misleading information. As shown in Figure 5, our model achieves the best results on both the H@1 and MRR metrics with different entity image similarity settings. We believe that our model can effectively leverage IB principle in training, which encourages the model to focus more on the task-relevant image information for prediction rather than the overall redundant image information.

*4.4.3 Performance on samples with high-noise image.* To explore the efficacy of entity alignment in noisy data scenarios, we utilize FB15K-DB15K dataset (20% seed alignments) and add artificial noise to the images for experiments. Specifically, we first obtain the entity's image embedding from the pre-trained visual encoder, and then impose random dropout to them with a dropout rate to control the noise rate. As depicted in Figure 6, results indicate that all models reflect a performance decline with increasing the image

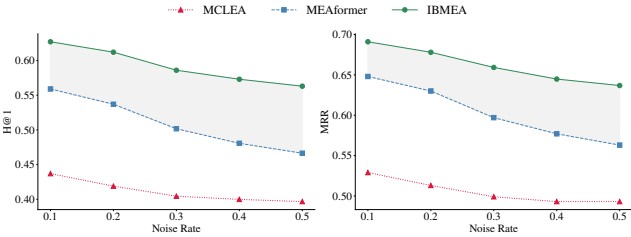

**Figure 6: Results on the samples with noisy image in FB15K-DB15K dataset. We impose random dropout to images as noise and control the dropout rate for analysis.**

| Case Entity | MMKG-1 Image | MMKG-2 Image | ImgSim | Rank in Prediction | | |
|---|---|---|---|---|---|---|
| | | | | MCLEA | MEAformer | IBMEA |
| ① Esteban Granero | | | 0.76 | 14 | 13 | 1 |
| ② List of counties in Oregon | | | 0.74 | 9 | 13 | 1 |
| ③ Run Run Shaw | | | 0.57 | 12 | 46 | 1 |

**Figure 7: Case study on entities with image similarity.**

noise rate. However, even under a higher noise rate, our model maintains relatively high H@1 and MRR metrics. Considering the higher noise rate tends to have more misleading clues, the results demonstrate the effectiveness of our model with IB regularizers, which successfully preserves alignment-relevant information while suppressing alignment-irrelevant information, thereby sustaining superior alignment performance in noisy environments.

*4.4.4 Case study.* To detail the model prediction, we present typical cases of aligned entities from DBP15K$_{ZH-EN}$ with relatively low-similarity figures in Figure 7. Case 1 shows characters with different clothes but similar facial features. Case 2 displays maps with different background colors yet identical contour features. In Case 3, a statue and a black-and-white photo of the same person exhibit similar facial traits, providing crucial information for entity alignment. The three cases reflect misleading clues in images. As in the predictions, it was found that MCLEA and MEAformer predict the true entity with a lower ranking. Note that in Case 3, the MEAformer model predicts worse, likely due to being misled by differences in entity forms. In contrast, our model accurately predicted in all cases, thus demonstrating its superior ability to handle misleading information for the MMEA task.

## 5 Related work

### 5.1 Entity Alignment

Entity Alignment (EA) aims to identify equivalent entities across different knowledge graphs (KGs) to facilitate knowledge fusion. The majority of existing EA methods focus on traditional knowledge graphs can be classified into two categories: 1) *Structure-based* techniques focus on employing knowledge embedding [1, 7, 12, 33, 54] to capture the entity structure information from relational triples, or utilize graph-based models [18, 27, 34, 45] for neighborhood entity feature aggregation [17, 21, 30, 40]. 2) *Side information-based* methods [2, 25, 32, 37, 46] integrate side information (e.g., entity name, entity attribute, relation predicates) to learn more informative entity representation. As previous studies [25, 51] have shown, the *structure-based* techniques assume that aligned entities should share similar neighborhoods, the *side information-based* methods regard equivalent entities usually contain similar side information. Nevertheless, these methods mainly solely on structural information and utilize textual side information, making them incapable of handling the visual data of entities in actual scenes.

### 5.2 Multi-Modal Entity Alignment

Due to the increasing popularity of multi-modal knowledge graphs, how to incorporate visual modalities in EA, i.e. multi-modal entity alignment (MMEA), has attracted research attention. The pioneer method PoE [24] defines overall probability distribution as the product of all uni-modal experts. Chen et al. [10] design a multi-modal fusion module for integrating different modal embeddings. HMEA [15] combines the structure and visual representations in hyperbolic space. MSNEA [11] and XGEA [47] integrate visual features to guide relational and attribute learning, with the former using a translation-based KG embedding method and the latter using a graph neural network (GNN) approach. ACK-MMEA [19] proposes a method for uniformizing multi-modal attributes, while UMAEA [13] introduces multi-scale modality hybrid and circularly missing modality imagination. PSNEA [28] and PCMEA [42] dynamically generate pseudo-label data to improve alignment performance. EVA [23] allows the alignment model to obtain the importance of different modality weights to fuse multi-modal information from KGs into a joint embedding. MCLEA [22] follows the EVA fusion method and then obtains informative entity representations based on contrastive learning. MEAformer [3] develops UMAEA and dynamically learns modality weights for each entity via the transformer-based attention fusion method. However, the above existing methods rarely explicitly address the misleading alignment-irrelevant information in each modality, resulting in incomplete utilization of the alignment-relevant information.

## 6 Conclusion

In this paper, we explore MMEA to identify equivalent entities between MMKGs. To address alignment-irrelevant misleading clues in modalities, we propose a novel MMEA framework termed as IBMEA, which aims to emphasize alignment-relevant information and simultaneously suppress alignment-irrelevant information in modalities. Particularly, we devise multi-modal variational encoders to represent multi-modal information with probability distributions and propose information bottleneck regularizers and an information contrastive regularizer. Experimental results indicate that our model outperforms previous state-of-the-art methods, and shows promising capability for misleading information alleviation, especially in the low-resource and high-noise data scenarios.

## Acknowledgments

This work was supported by the National Key Research and Development Program of China (Grant No.2021YFB3100600), the Youth Innovation Promotion Association of CAS (No.2021153), and the Postdoctoral Fellowship Program of CPSF (No.GZC20232968).

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
