# OpenReview forum: "IBMEA: Exploring Variational Information Bottleneck for Multi-modal Entity Alignment"
_acmmm.org/ACMMM/2024/Conference — MM2024 Poster_

### Official Review · Reviewer_Vs7k · 2024-05-23

**Rating:** 4
**Confidence:** 2

**Summary:**

This paper proposes a novel Variational Information Bottleneck Method (IBMEA) for Multimodal Entity Alignment (MMEA). MMEA aims to identify equivalent entities in different Multimodal Knowledge Graphs (MMKGs), which are typically associated with relevant images. Existing studies often fail to effectively suppress alignment-irrelevant information when fusing multimodal information, leading to subpar performance in low-resource or high-noise data scenarios. To address this issue, the paper suggests using the Information Bottleneck to emphasize alignment-relevant information and suppress irrelevant information, thereby generating entity representations. Specifically, the method includes designing a multimodal variational encoder to generate modality-specific entity representations and proposes four modality-specific information bottleneck regularization methods and a modality-mixed information contrastive regularization method.

**Strengths:**

1. IBMEA is the first to apply the Information Bottleneck (IB) to multimodal entity alignment, providing a new perspective to address the problem of misleading information during alignment.
2. The paper introduces two types of regularization methods: modality-specific information bottleneck regularization and modality-mixed information contrastive regularization, which optimize modality-specific and mixed representations, respectively, enhancing the model's robustness against misleading information.
3. Extensive experiments on five benchmark datasets demonstrate that IBMEA surpasses existing state-of-the-art methods across various data settings, showing exceptional performance in low-resource and high-noise data scenarios.

**Limitations:**

1. Complexity: The application of the Information Bottleneck principle to multimodal entity alignment by IBMEA is very novel and promising, warranting further exploration of how this approach can be applied to other multimodal tasks.
2. Interpretability: While the proposed method performs excellently in experiments, the internal mechanisms of the model and the specific contributions of different regularization terms may not be easily interpretable, requiring further research to enhance the model's transparency.

**Suitability:**

3

---

### Official Review · Reviewer_oCQA · 2024-05-25

**Rating:** 5
**Confidence:** 2

**Summary:**

This paper proposes a variational information bottleneck for multi-modal entity alignment (IBMEA), suppressing redundant modality information between multi-modal knowledge graphs. It also proposes modal-specific information bottleneck regularizers to limit the misleading clues in refining modal-specific entity representations. Experimental results demonstrate that the model consistently outperforms previous state-of-the-art methods, and also shows promising and robust performance in low-resource and high-noise data scenarios.

**Strengths:**

1. This paper designs a novel variational framework with information bottleneck (IB) principle, explores IB to alleviate misleading clues in MMEA. The experimental section well demonstrates the strategy's effectiveness.
2. It proposes 2 regularizers to refine the modal-specific and modal-hybrid features, improving the robustness with misleading clues.
3. The paper is well written and organized.

**Limitations:**

1. I hope to see more experimental evidence demonstrating the effectiveness of this two kinds of regularizers.
2. It's lacking in novelty in utilizing IB to suppress redundant modality information.

**Suitability:**

3

---

### Official Review · Reviewer_Y7xC · 2024-05-27

**Rating:** 2
**Confidence:** 4

**Summary:**

The research paper discusses Multi-modal Entity Alignment (MMEA) in the context of multi-modal knowledge graphs (MMKGs), where entities are linked to related images. Traditional methods struggle with irrelevant and misleading information, particularly in data-poor or noisy conditions. To address this, the authors propose a novel framework called IBMEA, which uses the Information Bottleneck (IB) principle to enhance entity alignment by focusing on relevant information and filtering out irrelevant data. The framework employs multi-modal variational encoders and introduces two types of regularizers: modal-specific information bottleneck regularizers and a modal-hybrid information contrastive regularizer. These components work together to refine and integrate entity representations, improving alignment accuracy. Extensive experiments demonstrate that IBMEA surpasses existing methods, showing strong performance even in challenging scenarios.

**Strengths:**

1.The paper introduces a novel approach to Multi-modal Entity Alignment (MMEA) by leveraging the Information Bottleneck (IB) principle. This is the first known application of IB to alleviate the impact of misleading clues in MMEA, providing a fresh perspective on enhancing entity alignment through explicit emphasis on relevant information and suppression of irrelevant data.

2. The proposed IBMEA framework has significant applications in the field of knowledge graph alignment, particularly in scenarios involving multi-modal data. Its ability to handle low-resource and high-noise environments makes it a valuable tool for real-world applications where data quality and quantity can be challenging.

3. The paper provides extensive empirical evaluation on five benchmark datasets, including two cross-KG and three bilingual MMEA datasets. The experiments results demonstrated that IBMEA consistently outperforms state-of-the-art methods.

**Limitations:**

The main issues of this article lie in the following two aspects:

**1. Unclear Relationship Between Information Bottleneck Theory and Multi-modal Alignment**

1) The connection between information bottleneck (IB) theory and multi-modal entity alignment (MMEA) is not clearly established. For instance, Figure 1 does not effectively illustrate the application of IB in the context of MMEA. Actually, the image similarity can be enhanced by focusing on specific modal information, such as grayscale or texture images, which resembles the concept of background removal. However, this does not convincingly demonstrate the role of IB.

2) The paper introduces IB to address the representation and identification of alignment-relevant and irrelevant features, a common issue in multi-modal data fusion and entity alignment. However, the physical meaning of IB in the context of MMEA remains ambiguous. The paper should reference prior works, such as [33], to better define the problem and clarify the relevance of IB. Additionally, the rationale for using IB to solve these issues, and the specific attributes of IB that make it suitable for feature representation in MMEA, need further explanation. This ambiguity undermines the paper's motivation and makes it appear as if a general theory is being applied to a specific application without sufficient justification.

3) In the methodology section, the necessity of IB for MMEA tasks is not adequately summarized. The paper does not explain why IB is essential, nor does it provide the theoretical background supporting its use. The Modal-specific Information Bottleneck Regularizer appears to merely apply mutual information, and its theoretical connection to IB needs to be further clarified.

**2. Inconsistencies in Thesis Arguments**

1) The paper claims to focus on multi-modal entity alignment, but the bilingual datasets used in the experiments do not appear to include multi-modal data such as images. The source and usage of image data are not provided or explained. This discrepancy needs to be addressed to maintain consistency in the paper's arguments.

2) While the paper emphasizes the theoretical application of IB in MMEA, the experimental section lacks theoretical analysis and proof of effectiveness regarding the IB principle. There are no experiments specifically measuring the impact of IB. The paper should include a detailed theoretical analysis and experimental verification of how the IB principle is effectively utilized, which would strengthen the overall argument and contribution of the study.


To sum up, the motivation of this paper is interesting, and the application of the information bottleneck principle is relatively novel. However, the two main points discussed above require further clarification. Additionally, more extensive effectiveness analysis is necessary, such as measurement experiments of information bottlenecks in MMEA tasks. In my opinion, the paper requires major revisions. I hope my comments will be helpful to the authors.

**Suitability:**

3

---

### Meta-Review · Area_Chair_65RW · 2024-06-26

**Recommendation:** Accept (Poster)
**Confidence:** 4

**Metareview:**

I have went through the entire process, including the submission, responses and discussion phases. The reviewers raised up some concerns to the authors, while the authors made the clarifications. However, the reviewers are NOT posing the final justifications and comments/discussions based on the authors' responses. I encouraged the authors to take the comments into considerations for their final version, such as clarifying some misunderstandings. I hence recommend the acceptance as a poster.